# Affective Temperaments and Clinical Course of Bipolar Disorder: An Exploratory Study of Differences among Patients with and without a History of Violent Suicide Attempts

**DOI:** 10.3390/medicina55070390

**Published:** 2019-07-19

**Authors:** Giovanna Fico, Vito Caivano, Francesca Zinno, Marco Carfagno, Luca Steardo, Gaia Sampogna, Mario Luciano, Andrea Fiorillo

**Affiliations:** 1Department of Psychiatry, University of Campania “L. Vanvitelli”, Largo Madonna Delle Grazie, 80139 Naples, Italy; 2Department of Health Sciences, Psychiatric Unit, University Magna Graecia of Catanzaro, Viale Europa, 88100 Catanzaro CZ, Italy

**Keywords:** bipolar, suicide, affective temperament, violent suicide, aggressive behaviors

## Abstract

*Background and Objectives:* Suicide is the leading cause of death in patients with Bipolar Disorder (BD). In particular, the high mortality rate is due to violent suicide attempts. Several risk factors associated with suicide attempts in patients with BD have been identified. Affective temperaments are associated with suicidal risk, but their predictive role is still understudied. The aim of this study is to assess the relationship between affective temperaments and personal history of violent suicide attempts. *Materials and Methods*: 74 patients with Bipolar Disorder type I (BD-I) or II (BD-II) were included. All patients filled in the short version of Munster Temperament Evaluation of the Memphis, Pisa, Paris and San Diego (short TEMPS-M) and the Temperament and Character Inventory, revised version (TCI-R). The sample was divided into two groups on the basis of a positive history for suicidal attempts and the suicidal group was further divided into two subgroups according to violent suicide attempts. *Results*: Violent suicide attempts were positively associated with the cyclothymic temperament and inversely to the hyperthymic one. BD-I patients and patients with a clinical history of rapid cycling were significantly more represented in the group of patients with a history of violent suicide attempts. *Conclusions*: Our study highlights that several clinical and temperamental characteristics are associated with violent suicide attempts, suggesting the importance of affective temperaments in the clinical management of patients with BPI.

## 1. Introduction

According to the World Health Organization (WHO), suicide is one of the leading causes of death worldwide accounting for nearly 800,000 deaths every year [1]. Psychiatric disorders are an important contributing factor to suicide attempts [2] and bipolar disorders (BDs) are associated with the highest suicide risk [3], which is 15–30 times higher than the general population. Up to one-half of patients with BDs attempt suicide in their lifetime [4]. 

Suicide is a complex and multivariate phenomenon, defined as “death caused by self-directed injurious behavior with intent to die” [5] which can be performed by several means, including violent and non-violent methods. Suicide methods are noteworthy to be characterized, since they can contribute defining a suicidal subpopulation more vulnerable to suicide completion. In fact, violent suicide attempters have been proved to have an elevated risk of future suicide attempts [6], as well as a higher immediate lethality of an attempt [7] with a fatality rate over 70% [8]. Although a violent suicide attempt is usually defined by the method, there is no clear consensus about its definition. Asberg [9] identifies a few violent attempts (i.e., hanging, gas poisoning and drowning), whereas drug overdoses are considered to be non-violent suicide attempts. Subsequently, Giner (2014) [10] and Penas-Lledo (2015) [11] adopted and extended Asberg’s criteria by including in the definition of “violent attempts” the use of firearms, jumping from heights, several deep cuts, car crash, burning, electrocution, and jumping under a train. Likewise, Dumais (2005) [12] enlarges the list of non-violent suicide attempts by adding drowning and gas poisoning. The identification of a subpopulation, such as the violent suicide attempters, of patients with mental disorders at higher risk to commit suicide and the identification of the risk factors of suicide attempts is a stepping stone in the development of a better suicide prevention strategy. 

Several risk factors linked to suicide attempts in patients with BD have been identified in previous studies, including a long duration of illness, untreated BD, female sex, positive history for suicide attempts, comorbidity with substance abuse or personality disorders, anxiety, depressive polarity [13], recent affective episodes, and recent psychiatric inpatient care [14]. Personality and temperamental traits have also been considered, and anger, impulsivity, aggression, anxiety and two factors (i.e., “harm avoidance” and “novelty seeking”) of the Temperament and Character Inventory (TCI) have been identified to be associated with suicide attempts [15]. Only recently temperament and, in particular, affective temperaments have been considered as possible factors linked to suicide attempts [16]. Temperament represents the “temporally stable biological core of personality” affecting several aspects of an individual’s life (activity level, rhythms, moods and related cognitions) while personality, a broader phenotype, also refers to “acquired characterological determinants and interpersonal operations” [16]. Akiskal et al. conceptualized a *spectrum* of *affective* conditions ranging from *temperament* to clinical episodes [17] and proposed criteria defining five temperaments: (1) Cyclothymic temperament, characterized by chronic cycling between mood polarities and unstable self-esteem and energy; (2) hyperthymic temperament, characterized by increased energy and optimism [17]; (3) irritable temperament, characterized by irritable and angry behavior; (4) anxious temperament, characterized by a tendency to worry; (5) depressive temperament, characterized by low levels of energy, introversion and worrying [18]. Affective temperaments, conceptualized as stable, subclinical forms of the manic-depressive illness by Akiskal (1983), have a role in the clinical evolution of the mood disorders and the outcome, including the risk of suicide attempts [19]. In particular, hyperthymic temperament has been shown to be associated with a reduced risk of suicide attempts [20,21], whereas cyclothymic, irritable, depressive and anxious temperaments are more present in patients with a positive history of suicide attempts [22,23]. Besides, cyclothymic and irritable temperaments are highly connected with both aggression [24] and impulsivity [25], which play a role in suicidal behavior. A widely used questionnaire to evaluate affective temperaments is the Temperament Evaluation of Memphis, Pisa, Paris and San Diego (TEMPS) [18,26] which has been developed in different formats and used in clinical and research settings [27]. It has been documented that affective temperaments individually considered, have reduced predictive power if used to anticipate suicidal risk [28], while composite TEMPS-A score yields stronger associations with suicidal risk and better identify subjects at risk for suicide attempts [20]. A few data on the clinical characteristics of violent attempters with BDs are available. Existing data suggest that violent attempters are more likely to be men [29], have the first episode of manic/hypomanic type [30] and carry the serotonin transporter gene S allele [31]. 

To our knowledge, there are no previous studies that specifically addressed the issue of the association between affective temperament subtypes and a history of violent suicide attempts in BDs. Thus, the aim of our study is to investigate the relationship between affective temperaments and personal history of violent suicide attempts, defined according to Asberg’s criteria, in a clinical sample of bipolar patients. 

## 2. Materials and Methods

The exploratory study was carried out in the bipolar outpatient unit of the Department of Psychiatry of the University of Campania “Luigi Vanvitelli”, in Naples between January and June 2018. The only inclusion criterion was a diagnosis of bipolar disorder type I or II according to the DSM 5 criteria. Patients were excluded if they have presented affective illness as a consequence of alcohol/substance abuse or dependence, medical illness, an organic brain disorder or medication, and if not able to provide written informed consent (i.e., dementia, cognitive impairment or delirium). The study was approved by the local ethical review board (N001567/28.01.2018). 

All patients filled in the short version of the Munster Temperament Evaluation of the Memphis, Pisa, Paris and San Diego (short TEMPS-M) [32,33], a 35 items questionnaire used to assess affective temperaments described by Akiskal (depressive, anxious, hyperthymic, cyclothymic and irritable) using a dimensional approach with a five-point Likert type scale ranging from 1 to 5 (1 = “not at all”; 2 = “a little”; 3 = “moderately”; 4 = “much”; 5 “very much”) [34] and the Temperament and Character Inventory, revised version (TCI-R) [35]. The TCI was developed by Cloninger, with the goal of assessing factors underlying the psychobiological aspect of personality. The revised version (TCI-R) is a questionnaire consisting of 240 items, with a 5-point Likert-type scale, grouped into four temperament dimensions, novelty seeking (NS), harm avoidance (HA), reward dependence (RD) and persistence (PS), and three character dimensions, self-directiveness (SD), cooperativeness (CO) and self-transcendence (ST) samples [36]. 

Patients’ and clinicians’ socio-demographic and clinical characteristics were analyzed using descriptive and frequency counts, as appropriate. The sample was divided into two groups on the basis of a positive history for a suicidal attempt. Differences among groups were evaluated using χ^2^ or Bonferroni-adjusted T-test, as appropriate. Suicidal patients were further divided into two subgroups according to the history of violent suicide attempts, in line with Asberg’s criteria (1976). A logistic multivariable regression model was performed in order to identify factors associated with a positive history of suicidal behavior. All possible confounders were entered in the model. Statistical analyses were performed using SPSS version 18, the level of statistical significance was set at the level of *p* < 0.05. 

## 3. Results

The sample consists of 74 patients. Patients’ sociodemographic characteristics, as well as patients’ clinical features, are reported in Table 1. The sample is composed by 36 men (48.7%) and 38 women (51.3%), with a mean age of 48.92 years (SD = 12.38), the most frequent diagnosis was bipolar II disorder (59.5%). The samples grouped according to a positive history of suicide attempts (Table 2). Forty patients have a positive history of suicide attempts (45.9%). Psychotic symptoms are more frequent in patients with a history of suicide attempts compared to patients without a history of suicide attempts (*p* < 0.019), and particularly during mixed episodes (*p* < 0.008). Furthermore, suicidal attempters show a higher rate of aggressive behaviors, but no differences were found between the two groups in terms of the clinical course of the disease, number of psychiatric admissions, number of affective episodes and seasonality. Depressive, anxious and cyclothymic temperaments are more represented in suicidal attempters, while hyperthymic in non-suicidal ones. Suicidal attempters have been further divided into two subgroups according to violent suicide attempts in line with Asberg’s criteria (Table 3). Twenty-four patients report a history of violent suicide attempts, 54.2% being females and 45.8% males with a mean age of 49.46 (SD = 13.02); the most frequent diagnosis is BDI (75%); 41.7% of violent attempters and 45.83% of non-violent attempters were on lithium therapy. A clinical course of rapid cycling (4 or more affective episodes during a year) is significantly more represented in the group of patients with a history of violent suicide attempts. Furthermore, violent attempters show a higher rate of cyclothymic temperament and lower rates of hyperthymic temperament. At the multivariate logistic regression, we found that the hyperthymic temperament reduces the likelihood to have a positive history of suicidal attempts (*p* < 0.01) (Table 4).

## 4. Discussion

The impact of affective temperaments on the risk of suicidal behavior is an emerging theme in the field of mental health [18,37,38]. Our sample is mainly composed of BD-II patients, considered at the highest risk of suicide attempts among patients with the bipolar spectrum [39]. Recent research showed that suicide attempts are 1.5 times more frequent among women than men and that the risk of suicide attempts decreases from BDII to major depression, BDI, other psychiatric disorders and to psychotic disorders, which carry the lowest risk [40]. This can be explained by the evidence that BDII patients experience more chronicity [41], higher rates of rapid cycling [42], greater disabling depressive symptoms [39], a higher probability of misdiagnosis [41], more anxiety disorders in comorbidity [43], agitated depression and residual symptoms [44]. Our study confirms that hyperthymic temperament is associated with a reduced risk of suicide attempts in BD patients, while depressive, cyclothymic and anxious temperaments have a strong association with suicide attempts [45]. These findings are in line with previous reports [16,22,46]. Hyperthymic temperament may exert protective effects in different ways, such as better drive, greater energy, more ambition, as well as better coping and decrease the risk of suicidal behavior [19,46]. On the other hand, the low rating of hyperthymic temperament has been associated with increased hopelessness, an important predictor of suicide attempts [47]. Contrary to previous studies [28], our data do not confirm the role of irritable temperament on suicide risk, probably due to the small sample size. In our study emerged that psychotic symptoms are more represented in patients with a history of suicide attempts particularly during mixed episodes [45]. Several studies have suggested that individuals who experience psychotic symptoms have an increased sensitivity to stress, in terms of affective reactions to life events [48], as well as poorer coping skills [49], which may contribute to a greater risk of suicidal behavior when faced with acute life stressors. Other potential mechanisms may be the presence of shared risk factors for suicidal behavior and psychotic symptoms, including traumatic early life experiences, especially physical and sexual abuse [45], as reported by a recent meta-analysis by Ng et al. [50]. Therefore, mixed states, both with or without psychotic symptoms, are associated with an elevated risk of suicidal behavior, due to a greater proportion of time spent being depressed than patients without mixed episodes [51]. Furthermore, our study has outlined several clinical characteristics of violent suicide attempters. Rapid cycling patients were more likely to be violent attempters, as well as patients with BD-I and with cyclothymic temperament. Increased impulsivity, which is often a characteristic of cyclothymic temperament, is associated with a worse prognosis of BD [52,53], and with a more severe clinical course of the disease with history of rapid cycling, mixed episodes and substance abuse [54]. Our findings are in line with other studies showing that cyclothymic temperament is associated with a behavioral instability, increased sensitivity to a stressful event and higher levels of impulsivity [22,55,56,57], and a risk factor for suicide attempts [21]. Therefore, identifying those conditions such as cyclothymic temperament, a rapid cycling course or mixed episodes, more related to an impulsive dimension in patients with BD, could help to more easily typify a subpopulation of patients at risk of committing violent suicide attempts. 

The higher rates of cyclothymic dominant temperament among patients with BD have been widely shown in several studies [38,57]. There is still an open debate whether cyclothymic temperaments and cyclothymia as a psychiatric disorder have overlapping characteristics. In particular, cyclothymia has been conceptualized as the extreme of a cyclothymic temperament, characterized by high rates of impulsivity, emotional lability and mood swings [55]; it is estimated that up to one third of patients with cyclothymia have more possibilities to develop BD, particularly the type II [58]. A great number of patients are misclassified as bipolar, depressed or with a personality disorder instead of receiving a diagnosis of cyclothymia by the use of DSM-5 diagnostic criteria [55,59,60]; thus, is it possible that patients with a diagnosis of cyclothymia with BDs have been included in the study. A broader approach, considering not only a categorical diagnosis based on strict DSM-5 criteria but including temperamental dimension also, could help to better identify patients with cyclothymia and plan individualized treatments [50]. 

Underdiagnosis or misdiagnosis are relatively common in BD [61,62], and they are linked with the presence of milder symptoms not fitting a BD diagnosis, but falling into the “soft bipolar spectrum” definition [63]. In particular, Akiskal and Pinto (1999) [63] described two subtypes of bipolar spectrum not associated with manic or hypomanic state: bipolar II1/2 (depression superimposed on cyclothymic temperament) and bipolar IV (depression superimposed on hyperthymic temperament). A recent work by Goto and colleagues [64] proved that depression in those who have cyclothymic (bipolar II1/2) and hyperthymic temperament (bipolar IV) may predict bipolarity, giving validity to the bipolar II 1/2 and bipolar IV concepts; this is in line with our results on cyclothymic temperament which is highly represented in the total sample and significantly higher in violent attempters. 

It is important to emphasize that this was an exploratory study with several potential limitations. The small sample size reduces the statistical power of our findings, especially in the sub-analysis made in the population of violent suicide attempters. Furthermore, we could not assess some socio-demographical characteristics and levels of education. Moreover, we only collected data on lithium treatment, but we have no data on other psychiatric medications. The time during which the study was conducted was relatively short; this can affect, for example, data on seasonality. Finally, data on aggression, as well as on clinical course, number or duration of admissions, presence of psychotic symptoms, were not assessed with objective measures, but they were referred by patients. The paper has also several strengths, such as the naturalistic setting and the fact that it is one of the first clinical studies on the possible role of affective temperaments in patients with BD. Moreover, in this study we evaluated the subpopulation of violent suicide attempters of BD patients, which can be considered an early detection target and can be taken into account for personalized treatments for BD. 

## 5. Conclusions

Our study outlined several clinical and temperamental characteristics of violent suicide attempters. Temperaments, in particular the affective ones, should be routinely assessed in clinical settings in order to identify people at higher risk of suicide attempts and to develop preventive programs, although it would be reductive to build a “suicide attempter profile” only on the basis of temperaments. A wider clinical evaluation, including also different clinical aspects such as current affective episodes and severity of the disease, pharmacological treatments, and the use of coping strategies in stressful situations should be taken into consideration in order to have effective early interventions. 

## Figures and Tables

**Table 1 medicina-55-00390-t001:** Socio-demographic and clinical characteristics of the sample (N = 74).

**Gender, male (N; %)**	(36; 48.6)
**Diagnosis, Bipolar disorder type I (N; %)**	(30; 40.5%)
**Age (mean ± SD)**	(48.92 ± 12.38)
**Age at illness onset (mean ± SD)**	(27.71 ± 9.8)
**Duration of illness (mean ± SD)**	(16.07 ± 9.46)
**Number of depressive episodes (mean ± SD)**	(7.1 ± 5.22)
**Number of manic episodes (mean ± SD)**	(3.79 ± 2.06)
**Number of hypomanic episodes (mean ± SD)**	(6.48 ± 4.5)
**Number of mixed episodes (mean ± SD)**	(2.53 ± 1.16)
**Lifetime number of affective episodes (mean ± SD)**	(14.17 ± 9.59)
**Number of affective episodes during last year (mean ± SD)**	(2.04 ± 2.28)
**Lifetime number of psychiatric admissions (mean ± SD)**	(0.493 ± 0.92)
**Duration of psychiatric admissions (mean ± SD)**	(0.7 ± 1.44)
**Clinical Course (N; %)**	
Mania–Depression–Interval (MDI)	(11; 14.9)
Depression–Mania–Interval (DMI)	(10; 13.5)
Mania–Interval–Depression (MID)	(8; 10.8)
Depression–Interval–Mania (DIM)	(3; 4.1)
Rapid cycling	(8; 10.8)
Irregular cycling	(25; 33.8)
**History of suicide attempts (N; %)**	(40; 54.1)
**Presence of psychotic symptoms, yes (N; %)**	(36; 49.3)
during depressive episodes	(13; 9.5)
during manic episodes	(7; 17.6)
during mixed episodes	(15; 20.3)
**Aggressive behaviours (N; %)**	(32; 43.2)
**brief TEMPS-M subscores (mean ± SD)**	
Depressive (dep)	(23.58 ± 5.93)
Cyclothymic (cyc)	(23.76 ± 7.1)
Irritable (irr)	(18.96 ± 7.59)
Anxious (anx)	(19.82 ± 6.3)
Hyperthymic (hyp)	(19.97 ± 6.3)

**Table 2 medicina-55-00390-t002:** Socio-demographic and clinical characteristics of the sample divided by history of suicide attempts.

Factors	Suicidal	Non-Suicidal
**Cases (N; %)**	40; 45.9%	34; 54.1%
**Gender, male (N; %)**	19; 47.5	17; 50
**Diagnosis, Bipolar disorder type I, (N; %)**	18; 45%	12; 30%
**Age (mean ± SD)**	49.53 ± 13.16	48.21 ± 11.56
**Age at illness onset (mean ± SD)**	26.76 ± 9.230	28.67 ± 10.45
**Duration of illness (mean ± SD)**	16.69 ± 8.92	15.34 ± 10.14
**Number of depressive episodes (mean ± SD)**	6.49 ± 3.55	7.82 ± 6.67
**Number of manic episodes (mean ± SD)**	3.93 ± 1.92	3.62 ± 2.24
**Number of hypomanic episodes (mean ± SD)**	5.61 ± 2.48	7.50 ± 5.96
**Number of mixed episodes (mean ± SD)**	2.55 ± 0.86	2.51 ± 1.45
**Lifetime number of affective episodes (mean ± SD)**	12.97 ± 5.92	15.59 ± 12.57
**Number of affective episodes during last year (mean ± SD)**	1.74 ± 0.68	2.39 ± 3.28
**Lifetime number of psychiatric admissions (mean ± SD)**	0.58 ± 1.07	0.38 ± 0.69
**Duration of psychiatric admissions (mean ± SD)**	0.800 ± 1.63	0.588 ± 1.2090
**Clinical Course (N; %)**		
Mania–Depression–Interval (MDI)	6; 15	5; 14.7
Depression–Mania–Interval (DMI)	6; 15	4; 11.8
Mania–Interval–Depression (MID)	3; 7.5	5; 14.7
Depression–Interval–Mania (DIM)	2; 5	1; 2.9
Rapid cycling	2; 5	6; 17.6
Irregular cycling	12; 30	13; 38.2
**Presence of psychotic symptoms, yes (N; %)**	25; 62.5 *	11; 32.4
during depressive episodes	4; 10	3; 8.8
during manic episodes	8; 20	5; 14.7
during mixed episodes	13; 32.5 *	2; 5.9
Seasonality (N; %)	16; 40.0	15; 32.4
Aggressive behaviours (N; %)	21; 52.5 *	11; 32.3
**brief TEMPS-M subscores** **(mean ± SD)**		
Depressive (dep)	25 ± 5.54	21.91 ± 6.01
Cyclothymic (cyc)	25.7 ± 6.38	21.47 ± 7.3
Irritable (irr)	19.68 ± 7.86	18.12 ± 7.29
Anxious (anx)	21.33 ± 6.31	18.06 ± 5.9
Hyperthymic (hyp)	18.13 ± 6.38 *	22.15 ± 5.53
**TCI-R subscores** **(mean ± SD)**		
NS total score	107.04 ± 18.19	106.34 ± 17.20
HA total score	113.85 ± 23.03	107.44 ± 23.87
RD total score	99.61 ± 16.54	99.04 ± 16.54
PS total score	100.09 ± 26.47	109.6 ± 24.85
SD total score	115.4 ± 22.16	123.2 ± 21.72
C total score	122.8 ± 16.14	124.3 ± 20.93
ST total score	73.13 ± 17.75	71.27 ± 13.25

Abbreviations: novelty seeking (NS), harm avoidance (HA), reward dependence (RD) and persistence (PS), self-directiveness (SD), cooperativeness (CO) and self-transcendence (ST). * *p* < 0.002 (Bonferroni).

**Table 3 medicina-55-00390-t003:** Socio-demographic and clinical characteristics of patients with a history of violent suicide attempts (N = 24).

**Gender, male (N; %)**	11; 45.8
**Diagnosis, Bipolar disorder type I, (N; %)**	18; 75%
**Age (mean ± SD)**	49.46 ± 13.02
**Age at illness onset (mean ± SD)**	27. 09 ± 8.73
**Duration of illness (mean ± SD)**	15.94 ± 10.42
**Number of depressive episodes (mean ± SD)**	7.57 ± 5.89
**Number of manic episodes (mean ± SD)**	3.76 ± 1.94
**Number of hypomanic episodes (mean ± SD)**	7.01 ± 5.20
**Number of mixed episodes (mean ± SD)**	2.52 ± 1.19
**Lifetime number of affective episodes (mean ± SD)**	14.88 ± 11.01
**Number of affective episodes during last year (mean ± SD)**	2.23 ± 11.01
**Lifetime number of psychiatric admissions (mean ± SD)**	0.380 ± 0 .69
**Duration of psychiatric admissions (mean ± SD)**	0.56 ± 1.12
**Clinical Course (N; %)**	
Mania–Depression–Interval (MDI)	4; 16.7
Depression–Mania–Interval (DMI)	5; 20.8
Mania–Interval–Depression (MID)	3; 12.5
Depression–Interval–Mania (DIM)	1; 4.2
Rapid cycling	8; 33.3 *
Irregular cycling	7; 29.2
**Presence of psychotic symptoms, yes (N; %)**	13; 54.2
during depressive episodes	3; 12.5
during manic episodes	6; 25
during mixed episodes	4; 16.7
**Aggressive behaviours (N; %)**	17; 70.8
**Lithium Therapy (N; %)**	10; 41.7%
**brief TEMPS-M subscores**	
Depressive (dep)	24.04 ± 6.29
Cyclothymic (cyc)	27.6 ± 5.39 *
Irritable (irr)	18.58 ± 7.64
Anxious (anx)	20.54 ± 5.71
Hyperthymic (hyp)	17.25 ± 5.60 *
**TCI-R subscores** **(mean ± SD)**	
NS total score	103.91 ± 18.59
HA total score	112.70 ± 26
RD total score	99.31 ± 13.9
PS total score	101.6 ± 27.45
SD total score	119.25 ± 21.79
C total score	124.02 ± 16.06
ST total score	71.3 ± 18.75

Abbreviations: novelty seeking (NS), harm avoidance (HA), reward dependence (RD) and persistence (PS), self-directiveness (SD), cooperativeness (CO) and self-transcendence (ST). * *p* < 0.002 (Bonferroni).

**Table 4 medicina-55-00390-t004:** Multivariate logistic regression, dependent variable: suicidal behavior vs. non-suicidal behavior.

**Number of subjects included on the analysis**	46		
**F (df)**	4.816 (1)		
**P**	0.000		
**Adjusted R square**	0.443		
**Constant**	1.572 (−2.037 to 2.493)		
	**O.R.**	**C.I. 95%**	***p***
**Lithium therapy**	0.185	0–9.628	0.149
**Age**	1.005	0.76–1.146	0.944
**Gender**	0.415	0.008–3.64	0.407
**Presence of psychotic symptoms**	0.692	0.003–7.39	0.726
**Aggressive behavior**	0.130	0.003–5.236	0.103
**Seasonality**	1.172	0.002–4.764	0.870
**Duration of illness**	1.026	0.862–1.215	0.668
**Lifetime number of affective episodes (1–10)**	-	0.852–1.356	0.229
**Lifetime number of affective episodes (11–21)**	11.233	0.523–1.112	0.098
**Lifetime number of affective episodes (>22)**	12.099	0.654–1.288	0.139
**Lifetime number of psychiatric admissions**	0.702	0.123–1.789	0.598
**brief TEMPS-M subscores**			
Depressive	0.875	0.448–1.096	0.294
Hyperthymic	0.800	0.526–1021	0.018
Anxious	1.098	0.681–1.428	0.461
Cyclothymic	1.127	0.974–2.015	0.271
Irritable	1.064	0.719–1.147	0.414

F-test; df, degree of freedom; C.I., confidence interval; O.R., Odds Ratio.

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
