# Peer review of "Affective Temperaments and Clinical Course of Bipolar Disorder: An Exploratory Study of Differences among Patients with and without a History of Violent Suicide Attempts"

_medicina, 2019, doi:10.3390/medicina55070390_

Round 1
Reviewer 1 Report
Dear authors
your study reports interesting results on the role of temperament in Bipolar Disorder, specifically focusing on sucidal attempts. I would suggest the following revisions in order to increase the liability of your work:
1) Methodology: due to the low sample size you have to better specify the sample size calculation criteria you used. Even if Pearson's r requirements for sample size are nont always clear I would suggest to consider to better specify your criteria (especially in the analysis for violent sucide subgroup). I I recommend Bonett and Wright (2000, Psychometrika) "Sample
size requirements for estimating ¨Pearson, Kendall and Spearman
correlations". Moreover I would appreciate a explicit power calculation for regression.
2) Language: a few costruct from temperament scale may be confused with diagnostic criteria or disorders (eg cyclotimic). Please better specifiy and revise the paper accordingly.
3) Discussion: I would suggest to better specifiy differences and similarities between your results and existing (recent) studies.
4) Limitations: Please better specify the limitations. The low sample size is affecting all the analyses, and there are many other biases (eg selection criteria; sociodemografic variables; types of pharmacological treatments; presence of psychotherapeutic or educational interventions; etc.).
I hope these suggestions may help you in improving your paper.
Author Response
Dear authors, your study reports interesting results on the role of temperament in Bipolar Disorder, specifically focusing on suicidal attempts. I would suggest the following revisions in order to increase the liability of your work.
We are grateful to the reviewer for his/her positive and encouraging comment.
Methodology: due to the low sample size you have to better specify the sample size calculation criteria you used. Even if Pearson's r requirements for sample size are not always clear I would suggest to consider to better specify your criteria (especially in the analysis for violent siucide subgroup). I recommend Bonett and Wright (2000, Psychometrika) "Sample size requirements for estimating ¨Pearson, Kendall and Spearman correlations". Moreover I would appreciate a explicit power calculation for regression.
Thank you for this comment. We are aware that since no sample size has been determined at the beginning of the study, this may have influenced the outcomes of the study. This is an exploratory, naturalistic and observational study conducted on patients’ clinical records; however, we will increase the number of subjects recruited in order to improve statistical power of the analyses. For this reason, we have slightly modified the title adding the definition of “exploratory study”, we have specified it in the methods section and we have acknowledged it as a limitation of the study. Furthermore, as reported by Bonett and Wright, Pearson’s correlations is recommended only for sample sizes >25. Therefore, correlation analyses have been excluded from the revised version of the paper. Moreover, we have provided more data on regression model in Table 4.
Language: a few costruct from temperament scale may be confused with diagnostic criteria or disorders (eg cyclotimic). Please better specify and revise the paper accordingly.
Thank you for this comment. We have provided a clearer definition of cyclothymic temperament in the Introduction (lines 70-76) and we have specified when we are talking about cyclothymic temperament throughout the text. A paragraph on cyclothymia has been added according to the reviewer’s suggestion (lines 206-217).
Discussion: I would suggest to better specifiy differences and similarities between your results and existing (recent) studies.
We are very grateful for this comment. In the Discussion we have add more evidence on latest research on affective temperaments and bipolar disorders (lines 170-180; 206-228).
Limitations: Please better specify the limitations. The low sample size is affecting all the analyses, and there are many other biases (eg selection criteria; sociodemografic variables; types of pharmacological treatments; presence of psychotherapeutic or educational interventions; etc.). I hope these suggestions may help you in improving your paper.
Thank you for this comment. We have amended the limitations’ section according to your suggestions (lines 229-237).
Reviewer 2 Report
An interesting paper with findings supporting papers that identify the risk of suicide in those with Bipolar Disorder (BD) and in particular those with BD type - II eg The proportion of suicide attempts and completed suicides in the general population is 35:1, but for people with bipolar disorder, the ratio is 3:1 ( Simon et al., 2007 ), which indicates a higher lethality of attempts in BD patients. The majority of the studies show that BD patients in general, and BD-II subjects in particular, carry the highest risk of suicide. The findings that both agitated depression and anxiety disorder comorbidities are more frequent in BD-II than in BD-I, can explain the extremely high suicide risk of this subgroup ( Rihmer et al., 2001;Henry et al., 2003;Benazzi et al., 2004 ). Ref: Plans, L., Barrot, C., Nieto, E., Rios, J., Schulze, T. G., Papiol, S., . . . Benabarre, A. (2019). Association between completed suicide and bipolar disorder: A systematic review of the literature. Journal of Affective Disorders, 242, 111-122. doi:10.1016/j.jad.2018.08.054
There is considerable evidence that patients diagnosed with BD are at increased risk of suicidal behaviors compared both to the general population and those with other psychiatric disorders. The proposed integrated model builds on the BSM, in which the appraisal system is integral to the emergence of suicidal ideation via feelings of defeat and entrapment, and facilitated by motivational and volitional factors. Ref: Malhi, G. S., Outhred, T., Das, P., Morris, G., Hamilton, A., & Mannie, Z. (2018). Modeling suicide in bipolar disorders. Bipolar Disorders, 20(4), 334-348. doi:10.1111/bdi.12622
These include affective episodes and subtype of the bipolar disorder. We found that recent affective episodes, especially depressive episodes, were significant predictors of suicide. This is in line with previous studies suggesting that depressive 17 but not manic 32 episodes are associated with increased likelihood of suicide in bipolar disorder. Ref: Hansson, C., Joas, E., Pålsson, E., Hawton, K., Runeson, B., Landén, M., . . . Sahlgrenska Academy. (2018). Risk factors for suicide in bipolar disorder: A cohort study of 12 850 patients. Acta Psychiatrica Scandinavica, 138(5), 456-463. doi:10.1111/acps.12946
Despite the similarities (and I may be misreading the data), it seems, as a population this study group is different or it may be the model of care adopted that the duration of admission seems lower than expected as is the number of admissions. It would be useful for the authors to address this aspect.
The next area for comment relates to terminology. The authors often use the term violent suicide or history of violent suicide eg 24 patients report a history of violent suicide. I assume the authors mean it was a violent attempt rather than an actual suicide. The authors should clarify this aspect.
The authors comment in the discussion that 'Our study confirms the role of hyperthymic temperament in preventing suicide among BD patients ...'. I would suggest this comment be modified as I have not seen the evidence that hyperthymic is preventative, rather it is associated with a reduced risk of suicide.
It would be useful for the authors to consider the structure of paper with some paragraphs occupying half a page (though it may be the manner the document was downloaded).
Author Response
An interesting paper with findings supporting papers that identify the risk of suicide in those with Bipolar Disorder (BD) and in particular those with BD type - II eg The proportion of suicide attempts and completed suicides in the general population is 35:1, but for people with bipolar disorder, the ratio is 3:1 ( Simon et al., 2007 ), which indicates a higher lethality of attempts in BD patients. The majority of the studies show that BD patients in general, and BD-II subjects in particular, carry the highest risk of suicide. The findings that both agitated depression and anxiety disorder comorbidities are more frequent in BD-II than in BD-I, can explain the extremely high suicide risk of this subgroup ( Rihmer et al., 2001;Henry et al., 2003;Benazzi et al., 2004 ). Ref: Plans, L., Barrot, C., Nieto, E., Rios, J., Schulze, T. G., Papiol, S., . . . Benabarre, A. (2019). Association between completed suicide and bipolar disorder: A systematic review of the literature. Journal of Affective Disorders, 242, 111-122. doi:10.1016/j.jad.2018.08.054. There is considerable evidence that patients diagnosed with BD are at increased risk of suicidal behaviors compared both to the general population and those with other psychiatric disorders. The proposed integrated model builds on the BSM, in which the appraisal system is integral to the emergence of suicidal ideation via feelings of defeat and entrapment, and facilitated by motivational and volitional factors. Ref: Malhi, G. S., Outhred, T., Das, P., Morris, G., Hamilton, A., & Mannie, Z. (2018). Modeling suicide in bipolar disorders. Bipolar Disorders, 20(4), 334-348. doi:10.1111/bdi.12622. These include affective episodes and subtype of the bipolar disorder. We found that recent affective episodes, especially depressive episodes, were significant predictors of suicide. This is in line with previous studies suggesting that depressive 17 but not manic 32 episodes are associated with increased likelihood of suicide in bipolar disorder. Ref: Hansson, C., Joas, E., Pålsson, E., Hawton, K., Runeson, B., Landén, M., . . . Sahlgrenska Academy. (2018). Risk factors for suicide in bipolar disorder: A cohort study of 12 850 patients. Acta Psychiatrica Scandinavica, 138(5), 456-463. doi:10.1111/acps.12946. Despite the similarities (and I may be misreading the data), it seems, as a population this study group is different or it may be the model of care adopted that the duration of admission seems lower than expected as is the number of admissions. It would be useful for the authors to address this aspect.
We are particularly grateful to the reviewer for this comment and to have raised this important issue. We have added a whole paragraph about suicidal risk in patients with bipolar II disorder from line 170 to 180. Our results on the sample are in line with recent findings on suicide risk, comorbidities and role of affective temperament in BDs. Unfortunately, we could not relate the number or duration of hospitalizations to clinical outcomes as we do not have such data.
The next area for comment relates to terminology. The authors often use the term violent suicide or history of violent suicide eg 24 patients report a history of violent suicide. I assume the authors mean it was a violent attempt rather than an actual suicide. The authors should clarify this aspect.
Thank you for your comment. The text has been modified accordingly.
The authors comment in the discussion that 'Our study confirms the role of hyperthymic temperament in preventing suicide among BD patients ...'. I would suggest this comment be modified as I have not seen the evidence that hyperthymic is preventative, rather it is associated with a reduced risk of suicide.
Thank you for your comment. The text has been changed (lines 80 and line 178).
It would be useful for the authors to consider the structure of paper with some paragraphs occupying half a page (though it may be the manner the document was downloaded).
Thank you. In this new version, the paragraphs’ format is fixed.
Reviewer 3 Report
Although this study does have its merits, I have several concerns about its content and scientific soundness. These should be thoroughly addressed before publication can be advised. Specific comments: - Please change "has been proved" to "has been shown". - Akiskal has energetically described several types of the bipolar spectrum, such as bipolar 1/2 (schizobipolar disorder), bipolar I1/2 (depression with protracted hypomania), bipolar II1/2 (depression superimposed on cyclothymic temperament), bipolar III (repeated depression plus hypomania occurring solely in association with antidepressant or other somatic treatment), bipolar III1/2 (repeated hypomania occurring in the context of substance and/or alcohol abuse), bipolar IV (depression superimposed on the hyperthymic temperament) and so on. In this light, the results of this study perhaps supports the idea that cyclothymic and hyperthymic temperaments may predict bipolarity, and the validity of bipolar II1/2 and IV concept is supported. See: ncbi.nlm.nih.gov/pubmed/20699193. - "Cyclothymic" and "hyperthymic" temperaments should be clearly defined in the introduction to give readers a framework. - Please start a new paragraph from "To our knowledge, there are no previous studies that specifically". - Please change "have been proved" to "have been proven". - Did the authors seek Institutional Review Board (IRB) approval for the present study? Please provide the actual IRB study/approval number. - How was sample size determined? There is currently no evidence of power calculation. - Was the level of significance maintained at 0.05 for all t-tests? With multiple t-tests, the likelihood of incorrectly rejecting a null hypothesis (i.e., making a Type I error) increases. Bonferroni correction should have been done to compensate for this. - Besides prescribed medications, did these patients take any herbal or traditional supplements? St John's wort is a popular herbal supplement and hypericum-induced mania has been reported (citation: ncbi.nlm.nih.gov/pubmed/28064110). - When you write "41,7", do you mean 41.7%? Please correct this. - No sensitivity analyses were performed. - What N and percentage of nonviolent suiciders were on lithium therapy? This was not specified. - It is unclear if the authors assessed and adjusted for comorbid conditions that would confound the risk of suicidality. - There is still much controversy surrounding the diagnosis of cyclothymia as a mood disorder as opposed to a personality trait/disorder. The issue of cyclothymia as a disorder versus a personality or Axis II problem is one that requires more research. Furthermore, data emerging from both academic centers and from public and private outpatient facilities indicate that from 20% to 50% of all subjects that seek help for mood, anxiety, impulsive and addictive disorders turn out, after careful screening, to be affected by cyclothymia (citation: ncbi.nlm.nih.gov/pubmed/26005206). The proportion of patients who can be classified as cyclothymic rises significantly if the diagnostic rules proposed by the DSM-5 are reconsidered and a broader approach is adopted. Unlike the DSM-5 definition based on the recurrence of low-grade hypomanic and depressive symptoms, cyclothymia is best identified as an exaggeration of cyclothymic temperament (basic mood and emotional instability) with early onset and extreme mood reactivity linked with interpersonal and separation sensitivity, frequent mixed features during depressive states, the dark side of hypomanic symptoms, multiple comorbidities, and a high risk of impulsive and suicidal behavior. In other words, could the subjects in this study be misdiagnosed as bipolar disorder instead of, more rightly, cyclothymia. These issues should be discussed.Author Response
Although this study does have its merits, I have several concerns about its content and scientific soundness. These should be thoroughly addressed before publication can be advised.
We are grateful to the reviewer for his/her positive and encouraging comments.
Specific comments:
Please change "has been proved" to "has been shown".
Thank you; the sentence has been changed.
Akiskal has energetically described several types of the bipolar spectrum, such as bipolar 1/2 (schizobipolar disorder), bipolar I1/2 (depression with protracted hypomania), bipolar II1/2 (depression superimposed on cyclothymic temperament), bipolar III (repeated depression plus hypomania occurring solely in association with antidepressant or other somatic treatment), bipolar III1/2 (repeated hypomania occurring in the context of substance and/or alcohol abuse), bipolar IV (depression superimposed on the hyperthymic temperament) and so on. In this light, the results of this study perhaps supports the idea that cyclothymic and hyperthymic temperaments may predict bipolarity, and the validity of bipolar II1/2 and IV concept is supported. See: ncbi.nlm.nih.gov/pubmed/20699193.
We are particularly grateful to the reviewer for this comment, because it gives us the opportunity to deepen the soft bipolar spectrum concept and to relate it to our results. The new paragraph is from line 219 to 228.
"Cyclothymic" and "hyperthymic" temperaments should be clearly defined in the introduction to give readers a framework.
Thank you for this comment. A brief description of affective temperaments, including cyclothymic and hyperthymic ones, as conceptualized by Akiskal, has been added from lines 70 to 76.
Please start a new paragraph from "To our knowledge, there are no previous studies that specifically".
As suggested, a new paragraph started from line 92.
Please change "have been proved" to "have been proven".
Thank you. We have modified it accordingly.
Did the authors seek Institutional Review Board (IRB) approval for the present study? Please provide the actual IRB study/approval number.
Thank you for this comment. The study was approved by the Ethical Review Board of the University of Campania “L. Vanvitelli (N001567)”.
How was sample size determined? There is currently no evidence of power calculation.
Was the level of significance maintained at 0.05 for all t-tests?
The sample size was not determined at the start of the study, since this is an exploratory, naturalistic and observational study conducted on patients’ clinical records. We are planning to increase the number of subjects recruited in order to improve the statistical power of our analyses. Based on your comment, we have modified the title (by adding the definition of “exploratory study”), we have specified it in the methods section and we have acknowledged it among the study limitations. The level of significance was set at 0.05, and for Bonferroni adjusted t-test at 0.002.
With multiple t-tests, the likelihood of incorrectly rejecting a null hypothesis (i.e., making a Type I error) increases. Bonferroni correction should have been done to compensate for this.
We are grateful to the reviewer for this comment. We have repeated the multiple t-test with Bonferroni correction (p<0.002).
Besides prescribed medications, did these patients take any herbal or traditional supplements? St John's wort is a popular herbal supplement and hypericum-induced mania has been reported (citation: ncbi.nlm.nih.gov/pubmed/28064110).
We are grateful to the reviewer for this comment. All patients were under a psychopharmacological treatment regimen; patients who experienced affective symptoms (such as mania/hypomania or depression) due to medical conditions or other medications or supplements have been excluded (lines 102-105).
When you write "41,7", do you mean 41.7%? Please correct this.
Thank you. We have changed it to “41.7%” since it is a percentage (Tab. 3).
No sensitivity analyses were performed.
Data on sensitivity of the multivariate logistic regression have been added (Table 4).
What N and percentage of nonviolent suiciders were on lithium therapy? This was not specified.
The percentage of non-violent suicide attempters of our sample is 45.83% (N=11) (line 140).
It is unclear if the authors assessed and adjusted for comorbid conditions that would confound the risk of suicidality.
This information has been added in the exclusion criteria (lines 102-105).
There is still much controversy surrounding the diagnosis of cyclothymia as a mood disorder as opposed to a personality trait/disorder. The issue of cyclothymia as a disorder versus a personality or Axis II problem is one that requires more research. Furthermore, data emerging from both academic centers and from public and private outpatient facilities indicate that from 20% to 50% of all subjects that seek help for mood, anxiety, impulsive and addictive disorders turn out, after careful screening, to be affected by cyclothymia (citation: ncbi.nlm.nih.gov/pubmed/26005206). The proportion of patients who can be classified as cyclothymic rises significantly if the diagnostic rules proposed by the DSM-5 are reconsidered and a broader approach is adopted. Unlike the DSM-5 definition based on the recurrence of low-grade hypomanic and depressive symptoms, cyclothymia is best identified as an exaggeration of cyclothymic temperament (basic mood and emotional instability) with early onset and extreme mood reactivity linked with interpersonal and separation sensitivity, frequent mixed features during depressive states, the dark side of hypomanic symptoms, multiple comorbidities, and a high risk of impulsive and suicidal behavior. In other words, could the subjects in this study be misdiagnosed as bipolar disorder instead of, more rightly, cyclothymia. These issues should be discussed.
We are particularly grateful to the reviewer to have raised this important issue, which has been included in a new paragraph (lines 206-218).
Round 2
Reviewer 1 Report
Authors have significantly improved the paper.
Author Response
Thank you for your comment.
Reviewer 3 Report
Thank you for the revisions. Please see below for specific comments.
The authors use the word “suicidality”, which is no longer in favour today due to its imprecision. The authors also sometimes use suicide and suicide attempt interchangeably. Please standardize this.
Please change "The exploratory study has been carried out" to "The exploratory study was carried out".
"The study was approved by the local ethical review board." Please provide actual IRB study/approval number in the manuscript.
Please change "childhood traumatic experiences" to "traumatic early life experiences".
In addition to ref [45], authors should cite a recent meta-analysis, which showed that in both cross-sectional and cohort studies, early life sexual abuse was consistently associated with increased suicide attempts (citation: ncbi.nlm.nih.gov/pubmed/29454220). This is an important study confounder that was not investigated or adjusted for in your analysis.
Author Response
Thank you for the revisions. Please see below for specific comments.
The authors use the word “suicidality”, which is no longer in favour today due to its imprecision. The authors also sometimes use suicide and suicide attempt interchangeably. Please standardize this.
Thanks for your comment. The word “suicidality” has been replaced throughout the text with the words “suicide attempts” or “suicide risk”, depending on the context. Moreover, as suggested, the word “suicide” has been replaced with “suicide attempts”.
Please change "The exploratory study has been carried out" to "The exploratory study was carried out".
Thank you for this comment. The text has been changed accordingly (line 98)
"The study was approved by the local ethical review board." Please provide actual IRB study/approval number in the manuscript. Line 105
The IRB number has been added o line 104 of the manuscript.
Please change "childhood traumatic experiences" to "traumatic early life experiences".
Thank you. Done accordingly (line 193).
In addition to ref [45], authors should cite a recent meta-analysis, which showed that in both cross-sectional and cohort studies, early life sexual abuse was consistently associated with increased suicide attempts (citation: ncbi.nlm.nih.gov/pubmed/29454220). This is an important study confounder that was not investigated or adjusted for in your analysis.
We are particularly grateful to the reviewer for this comment, since it provides further evidence on the role of early life experiences as a risk factor for suicide attempts. We have added results from the suggested meta-analysis (lines 193-195).